# Thermal Analysis-Based Field Validation of the Deformation of a Recycled Base Course Made with Innovative Road Binder

**DOI:** 10.3390/ma14205925

**Published:** 2021-10-09

**Authors:** Grzegorz Mazurek, Przemysław Buczyński, Marek Iwański, Marcin Podsiadło

**Affiliations:** Department of Civil Engineering and Architecture, Kielce University of Technology, Al. Tysiąclecia Państwa Polskiego 7, 25-314 Kielce, Poland; p.buczynski@tu.kielce.pl (P.B.); iwanski@tu.kielce.pl (M.I.); mpodsiadlo@tu.kielce.pl (M.P.)

**Keywords:** recycled base, foamed bitumen, linear viscoelasticity, temperature distribution, field validation

## Abstract

The deformation of the cold recycled mixture with foamed bitumen in a recycled base with an innovative three-component road binder and foamed bitumen is analysed. Numerical simulation results for the pavement constructed, based on laboratory test results, were verified against the data from the monitoring system installed on the road trial section. In addition, environmental effects, such as air temperature and humidity levels in the pavement structure layers, were considered. Thermal analyses were conducted to identify the thermal properties of the pavement materials under steady heat transfer rate. Determining temperature distribution in the road cross-section in combination with relaxation functions determined for individual pavement layers contributed to the high effectiveness of the numerical simulation of deformation and displacement in the recycled base and the entire pavement. The experimental method of identifying thermal properties allows a fast and satisfactory prediction of temperature distribution in the pavement cross-section.

## 1. Introduction

Recycled mixtures are commonly used as a pavement base and subbase material in rehabilitated and new roads [1,2]. Although such mixtures with foamed bitumen have attracted growing interest from researchers [3,4,5], their linear viscoelastic behaviour, different from that of hot-mix asphalt (HMA) [6], must be extensively modelled.

The choice of an active binder substantially affects the physical and mechanical properties of cold-recycled mixtures with foamed bitumen (CRM-FB). According to Halles et al. [7], cement positively impacts a recycled mixture’s mechanical characteristics, such as stiffness and indirect tensile strength. The use of hydrated lime, in turn, increases the resistance to moisture-induced damage. A strong effect of high Ca(OH)_2_ content on a well-dispersed foamed bitumen resulted in strong bitumen structuring, as in the case of mastic reported in [8]. It must also be noted that slight changes in humidity result in major changes in mechanical characteristics, particularly in recycled mixtures with foamed bitumen [9,10,11]. According to the authors of [7], the use of cement and cement kiln dust (CKD) contributes to an increase in indirect tensile strength and stiffness modulus, whereas the fly ash acts as a fine filler. The research conducted by Iwański et al. [12] features the use of the experimental design that allows for seeking complex interactions in a three-component binder containing, among other things, fluidised bed combustion fly ash and hydrated lime. As a result, the researchers demonstrated that using the mixed binder improved the mixture’s water and frost resistance, thereby confirming its universal potential for providing the recycled mixture with an adequate structure. Each component in the binder plays a role of a catalyst for foamed bitumen dispersion or bitumen emulsion breakdown, leading to substantially improved mechanical parameters, such as stiffness modulus, indirect tensile strength, and fatigue resistance of recycled mixtures [3,13]. It must also be emphasised that the moisture resistance of recycled mixtures results from the interaction between cement and hydrated lime [10].

The presence of an active binder ensures the required cohesion of the CRM-FB and reduces the risk of exceeding the limit state [3,4]. However, the type of active binder determines the durability of recycled mixtures under cyclic loads. High cement content increases the stiffness of the mixture and effectively reduces the pavement’s fatigue life compared to hot-mix asphalt (HMA) [3,13]. Previously conducted research [12] revealed the beneficial effect of the optimum binder composition on the minimisation of premature thermally induced cracking. It should also be noted that an adequately designed binder will significantly modify the viscoelastic properties, thus helping to avoid deformations due to long-term loading. In [14], Mazurek et al. reported the results of a comprehensive creep analysis for the CRM-FB containing a binder mix composed of cement, hydrated lime and dust derived from de-dusting systems. They demonstrated that changes in the ratios of mixed binder components significantly modify the deformation rate in the recycled layer and the maximum depth of permanent deformation and that numerical methods can be used to predict these changes. It should be emphasised that the level of deformation is much lower than in HMA mixtures, as previously observed by Stimilli et al. [15].

Another important aspect strictly related to the behaviour of CRM-FBs is the correct identification of the stiffness change with respect to loading time and temperature [6]. A link between the binder’s effect and changes in the viscoelastic properties is presented in [16,17]. To reduce energy dissipation in the recycled mixture, it is important to design the system of structural layers so that the deformation at the base course lower part remains within the linear viscoelasticity range. It is worth noting that temperature has a substantial impact on the recycled mixture’s stiffness. Thermal sensitivity will strictly depend on the ratio: hydraulic binder-foamed bitumen-reclaimed asphalt. A precise analysis of deformation in the CRM-FB layer should therefore take into account the temperature distribution. There are many papers in which the temperature effect was taken into consideration in the stiffness modelling of specimens of recycled mixtures [17,18,19] and bituminous mixtures [20,21,22] using the approach applied in [23], or a more refined temperature distribution estimation method as in [24].

The literature lacks similar solutions that would utilise a three-component binder with a composition adjusted to local conditions. The influence of the binder on the recycled layer with rheological effects taken into account can be simulated with the use of numerical methods. However, when new materials are to be used, the numerical deformation results must be verified against the actual conditions through field testing on a trial road section. 

## 2. Materials and Methods

### 2.1. Binder Composition Optimisation

Composition of the binder used in the trial section was determined based on multi-criteria optimisation results. The testing plan followed the sampling regime consistent with the experimental design. The initial design of the mixture was modified by adding each component at an amount ranging from 20% to 80% (1/5 ÷ 2/5) [25,26]. Seven universal binder compositions were prepared and randomised. The binder compositions are presented in Table 1.

The sum of the mixture’s components could not be greater than 100%. More detailed information on the impact of the three-component binder on the standard mortar properties and chemical composition is available in [27,28]. 

The optimisation of the binder composition for the desired characteristics of the CRM-FB allowed the developing of the solution that ensured maximum fatigue life, high stiffness at high temperatures, and the lowest possible stiffness at low temperatures. Characteristic areas of the three-component binder composition solutions were established using the criteria above and the Harrington desirability function [26] to assess its suitability for use in the CRM-FB (Figure 1). 

Ultimately, the hydraulic binder used for the preparation of the recycled cold mixture with foamed bitumen was the binder with the composition corresponding to that identified as 5C (Figure 1b). The active binder denoted 5C was characterised by the following component ratios: 40% cement (CEM), 20% hydrated lime (CaOH_2_), and 40% dust derived from dust extraction systems (cement by-pass dust, CBPD). The result was a highly satisfactory solution [26] that met the maximum CBPD utilisation postulate. Furthermore, the quality of the 5C solution was similar to that containing 100% cement. The best solution was obtained for the area with 3V as a dominant binder. However, this solution can use only 20% CBPD, which makes it less useful for practical purposes. By way of a compromise, the composition denoted as 5C was ultimately chosen as the optimum solution. 

From the perspective of the optimisation surface, cement had the most powerful effect on the change in binder quality (Figure 1a). As shown in the Pareto chart, the positive correlation of Ca(OH)_2_ was also high. Although the CBPD component had a significant but the lowest influence on the optimisation solution, its interaction with hydrated lime appeared to be very strong. These results are consistent with the findings reported in [28]. According to this, hydrated lime acts as a stabiliser for the swelling due to the presence of GPBG in a blended binder. Hydrated lime was added to stabilise free bitumen in the recycled mixture and increase the stiffness of the mastic (a mixture of bitumen and mineral fillers <0.063 mm). Table 2 compiles the phase composition of the binder components. 

It follows from Table 2 that the cement has the phase composition that is typical of this material. In the case of CBPD, two phases exhibit binding properties: CaO/C2S and sylvine/calcite. All three components were required to be used and the optimisation solution did not allow the use of a two-component binder. 

### 2.2. Foamed Bitumen

Paving grade bitumen 70/100 was foamed and introduced directly into the mixer along with the mineral mixture containing the three-component binder.

Table 3 summarises the characteristics of bitumen 50/70 used for foaming. The data in Table 1 include an average confidence interval at the significance level α = 0.05.

The foaming was performed at 160 °C, 6.0 kPa water pressure and 5.5 kPa air pressure. The settings of the laboratory-scale foaming plant (WIRTGEN WBL10S) also included the earlier-developed model that bound the bitumen foaming parameters and chemical composition changes due to bitumen ageing during the foaming process [29]. The optimum foaming water content of 2.5% relative to the bitumen binder weight was established by using the results reported in [1,30] The optimum bitumen foam quality was characterised by maximum expansion E_max_ = 14.0 and half-life H-L = 10.0 s. The mixing time of the mineral mixture with the three-component binder was no longer than 180 s.

### 2.3. Recycled Mineral Mixture Design

The skeleton of the recycled mineral mixture was designed to include a high content of fine mineral particles. When foamed bitumen is used, fine particles ensure the correct formation of the mastic [31,32]. The CRM-FB mineral mixture gradation is presented in Figure 2.

The mineral skeleton consisted of 3.0% binder blend 5C, 56.4% natural aggregate 0/31.5 mm derived from the existing base course, mixed with 34.6% recycled asphalt (RAP 0/31.5 mm). Here, the RAP 0/31.5 mm was reclaimed by milling existing layers asphalt pavement. The binder content, 3% (m/m), in the CRM-FB corresponded to that of the laboratory mixtures.

### 2.4. Trial Section 

The design of the trial section included a monitoring system aimed to observe the changes in the stress and strain of the recycled base. In addition, environmental parameters, such as temperature and humidity were recorded. For this purpose, each measurement section was fitted with the following set of SHM System^®^ sensors:

System for monitoring the road section with linear fibre optic sensors:Linear optic-fibre strain sensors (EpsilonRebar), 8 m in length;Linear optic-fibre strain sensors (EpsilonRebar), 4 m in length;Linear optic-fibre displacement sensors (3D Sensor), 4 m in length.The system for monitoring the road section with point sensors (environmental factors) consisted of:Temperature sensors;Humidity sensors.

The sensors monitoring the geometrical changes of the pavement structure (displacements and strains) under vehicle load were embedded during the placement of each layer. Their innovative structure allowed them to be fully embedded in the base layer. Selected phases of the trial section preparation are presented in Figure 3.

In addition to the measurement of environmental factors, such as the temperature in the road structure, particularly in the CRM-FB_5C base course, the monitoring system recorded the temperature on the surface of the JENA type SMA 16 (SMA-JENA) layer and the soil at the depth of 30 cm. For comparison purposes, the recycled base on the adjacent lane of the trial section was constructed using the same procedure, but cement was used instead of the 5C binder. A scheme of the monitoring system sensor positioning is shown in Figure 4.

The basic task of this sensor layout was to record strains parallel to the road axis and strains along the road’s cross-section. Furthermore, additional displacement sensors were fitted in the soil below the CRM-FB_5C layer. The aim was to control displacement in the soil that could suggest exceeding boundary condition in the subgrade and the ability to estimate the actual fatigue life due to permanent deformation. On the other hand, the environmental sensors’ recording was aimed at enabling precise observance of the changes in the road cross-section’s temperature. It must be noted that the continuous environmental conditions monitoring system transmitted data, by using the GSM network protocols, to the external database in 15-min intervals. 

It must be noted that the paper focuses solely on the validation of the experimental results with the forecasted values of the CRM-FB_5C recycled base. Other layers, i.e., CRM-FB_cem (100% cement in the binder) and SMA-JENA constituted comparative materials and were required for conducting correct numerical analysis.

### 2.5. Heat Transfer in the Road Pavement

Heat transfer was an important element in the calculation of strain distribution on the road surface. The Fourier–Kirchhoff equation was used to describe heat conduction. A special case of heat transfer, defined as steady-state, was considered based on the literature data and own experience. It assumes that the temperature change in a given time interval (4 h) is constant or is sufficiently slow to make the change rate negligible. Accordingly, the heat flux density was directly proportional to the temperature [22]. The steady-state heat transfer can be expressed by a simplified Fourier Equation (1):(1)$$q=-\lambda \times \frac{\mathrm{dT}}{\mathrm{dy}}$$
where: q is heat flux density, W/m^2^; λ is thermal conductivity coefficient, W·m^−^^1^·K^−^^1^; and y is measurement ordinate in the pavement cross-section (depth), m. 

### 2.6. Stiffness Modulus

The dynamic modulus |E*| was tested using the DTC-CY method on cylindrical specimens to assess the rheological properties of CRM-FB mixtures with various binders. The test was conducted in accordance with PN-EN 12697-26D [33] and relied on cyclic axial loading of the cylindrical specimen. The test included recording the force, displacement, phase shift angle, and number of cycles, thereby enabling the calculation of the dynamic modulus |E*|. The following conditions were adopted for the dynamic modulus testing:Temperature: −10, 5, 13, 25, 40 °C;Frequency: 0.1, 0.3, 1, 3.5, 10, 20 Hz;Controlled strain 25 με;Load shape—sinusoid.

The detailed description of the stiffness test used was necessary since the dynamic modulus results were the basis for constructing the master curve using the time temperature superposition (TTSP) method.

### 2.7. Air Void Content, V_m_

Air void content (V_m_) is the volume of void space in the asphalt mixture (EN 12697-8). It plays a significant role in shaping the mixture properties. Insufficient content of air voids may reduce the resistance to permanent deformation. On the other hand, excessively high content in the mixture adversely affects its resistance to water and frost. The air void content in an asphalt mixture is determined from formula (2):(2)$$V_{m}=\frac{\rho_{m}-\rho_{b}}{\rho_{m}}\times 100\%$$
where

$V_{m}$ is air void content [0.1%];

$\rho_{m}$ is density of the asphalt mixture [Mg/m^3^];

$\rho_{b}$ is bulk density of the asphalt mixture [Mg/m^3^].

### 2.8. Indirect Tensile Strength

Indirect tensile strength was determined on specimens compacted as per EN 12697-30 with a Marshall hammer using 75 strokes per face (2 × 75) at 28 days of curing in air-dry conditions. The ITS test was carried out according to EN 12697-23 at a temperature of 25 °C (±1 °C). The strength value was calculated according to formula (3): (3)$$\mathrm{ITS}=\frac{2\times P}{\pi \times h\times d}$$
where:

ITS is indirect tensile strength [0.001 kPa];

P is maximum load at failure [kN];

h is specimen height to the nearest 0.1 mm [mm];

d is specimen diameter to the nearest 0.1 mm [mm].

### 2.9. Resistance to Water and Frost Damage, ITSR

The ITSR test was conducted according to EN 12697-12 [34] in compliance with the Polish Annex 1 to the technical requirements for bituminous mixtures, WT-2:2014 [35], containing application guidelines for uniform standards EN-13108-x. The specimens were conditioned in water and subjected to 1 cycle of freezing. The ITSR was determined 28 days after compaction; the strength tests were performed at a temperature of 25 °C (±1 °C); the result was calculated according to formula (4):(4)$$\mathrm{ITSR}=\frac{{\mathrm{ITSR}}_{\mathrm{WET}}}{{\mathrm{ITS}}_{\mathrm{DRY}}}$$
where:

ITSR is resistance to water and frost damage [%];

${\mathrm{ITSR}}_{\mathrm{WET}}$ is indirect tensile strength at 25 °C of conditioned specimens (kPa);

${\mathrm{ITS}}_{\mathrm{DRY}}$ is indirect tensile strength at 25 °C of non-conditioned specimens (kPa).

### 2.10. Uniaxial Compressive Strength, UCS

Uniaxial compressive strength (UCS) was determined on cylindrical specimens prepared with the Proctor method to the requirements of EN 13286-50. The test temperature was 25 °C ± 3 °C as per PN-EN 13286-41. The 28-day compressive strength was determined from formula (5).
(5)$$\mathrm{UCS}=\frac{F}{A_{C}}$$
where:

UCS is compressive strength of cement based specimens (N/mm^2^); F is maximum transferred force (N); A_C_ is cross-sectional area of the cement based specimens, (mm^2^).

### 2.11. Stiffness Modulus (Sm)

The stiffness modulus was determined to the requirements of the standard EN 12697-26 Annex C. The test consists of the measurement of the vertical and horizontal displacements at mid-height of the specimen and in the control of the force applied. The horizontal displacement was 5 ± 2 µm, and the diameter of the test specimens was 101.6 mm. The stiffness modulus can be calculated from formula (6):(6)$$Sm=\frac{F\times (\nu +0.27)}{z\times h}$$
where:

*Sm* is stiffness modulus of the specimen, [MPa];

F is maximum force applied to the specimen, [N];

*ν* is temperature-dependent Poisson’s ratio;

*z* is amplitude of the horizontal displacement of the specimen under loading, [mm];

ΔV is maximum vertical displacement of the specimen (corresponding to the maximum horizontal displacement), [mm].

### 2.12. Road Pavement Layer Viscoelasticity

Due to the presence of the bitumen binder, all bituminous composites demonstrate a dependency on the load time and the applied stress [36]. In relation to the above, it was required to adopt a suitable model that would take into consideration the rheological effects. Due to the fact that the base layers were loaded at approximately 10^−4^ m/m, the linear viscoelasticity range was adopted without taking into consideration the non-linear effects that depended on the stress level [8,36]. The general Maxwell model (GM) in the linear viscoelasticity (LVE) range was adopted as the model adequate for describing the material relaxation phenomenon [37]. 

There are many ways to determine the relaxation function. These include the static creep tests and dynamic tests with oscillating load conducted as a controlled strain in accordance with EN 12697-26 Annex D. The second phase of testing was substantially more adequate, because the instantaneous stiffness modulus E_o_ became the special target. In the case of static creep testing, there is no physical possibility of instantaneously loading the specimen. Furthermore, the correct specification of the relaxation function sequence using the stiffness modulus E(t) over time t→∞ would be ineffective. Due to the above, it was decided to use the dynamic test.

The parameters of the GM model can be obtained based on the measured imaginary (E’’) and real part (E’) of the complex modulus |E*| in the frequency domain by using the Fourier transform. Then, the component values of the dynamic modulus |E*| can be recorded with the following formula [38] (7):(7)$$E^{*}=\;E^{\prime }+{\mathrm{iE}}^{\prime \prime }$$
where: E’ is real part of the dynamic modulus E’’ is imaginary part of the dynamic modulus

The components of the dynamic modulus |E*| are represented by formulae (8) and (9)
(8)$$\;E\;\prime =E_{o}-\overset{N}{\underset{i=1}{\sum }}E_{i}+\overset{N}{\underset{i=1}{\sum }}\frac{E_{i}\times \tau_{i}^{2}\times \varpi_{i}^{2}}{1+\tau_{i}^{2}\times \varpi_{i}^{2}}$$
(9)$$\;E^{\prime \prime }=\overset{N}{\underset{i=1}{\sum }}\frac{E_{i}\times \tau_{i}\times \varpi_{i}}{1+\tau_{i}^{2}\times \varpi_{i}^{2}}$$
where: E’ is real part of the dynamic modulus, E’’ is imaginary part of the dynamic modulus, E_i_ is i-th elasticity modulus in the GM model, E_o_ is instantaneous elasticity modulus at t→0, τ_i_ is i-th relaxation time in the GM model, ω_i_ is reduced frequency.

The GM model parameters’ identification required using the non-linear least squares method. For this purpose, an original script for seeking the minimum goal functions was developed in the MathCad program with the Quasi-Newton method implemented. The obtained results were validated using the MCalibration^®^ software [39]. The tests were conducted for several temperature levels (−10 °C, 5 °C, 13 °C, 25 °C, 40 °C) aimed at representing the recycled mixtures’ behaviour to the greatest extent possible. Therefore, in order to include the temperature effect in the construction of the dynamic modulus master curve, it was required to use the classic time temperature superposition principle (TTSP) binding the load frequency and temperature in the form of a horizontal temperature shift factor $\alpha_{T}$. The Williams–Landel–Ferry (WLF) equation in the form was used for this purpose [40] (10).
(10)$${\log\alpha }_{T}=\frac{C_{1}\times \left(T-T_{0}\right)}{C_{2}+\left(T-T_{0}\right)}$$
where: C_1_, C_2_ are experimental factors, T_0_ is reference temperature, T is test temperature.

The quality of the model’s fit to the experimental data was determined using two qualitative measures, i.e.,: modified determination coefficient R^2^ and standardised root mean square error (RMSE) [37]. The master curve function was presented in a formula, taking into consideration the scaling in relation to the theoretical instantaneous stiffness modulus |E*( ω→∞)|. This was reasonable due to the fact that the tests were conducted at −10 °C with the load application frequency of 20 Hz. Due to the fact of using the ABAQUS program for the numerical calculations, the relaxation function was introduced by using the shear moduli of elasticity G and bulk moduli of elasticity K [41]. It was, therefore, necessary to define Poisson’s ratio, which was established as a constant value ν = 0.3. In effect, the relaxation function adopted for the modelling is represented by formulae (11) and (12):(11)$$G\left(t\right)=G_{0}-\left[\overset{n}{\underset{i=1}{\sum }}G_{i}\times \left[1-e^{\left(-\frac{t}{\tau_{i}}\right)}\right]\right]$$
(12)$$\mathrm{GG}=\frac{E}{2\left(1+\upsilon \right)}$$
where: $G_{0}$ is instantaneous shear modulus, $G_{i}$ is i-th shear modulus of stiffness (GM), $\tau_{i}$ is relaxation time, t is reduced load time.

## 3. Field Validation of CRM-FB Pavement Properties

### 3.1. Physical and Mechanical Properties

In order to verify and compare the physical and mechanical properties, selected tests of specimens prepared in the laboratory and collected from the CRM-FB layer based on the trial section were conducted. In addition, the results for CRM-FB_cem (100% cement) were used as a reference. The specimens collected from the trial section were collected as cores with 150 mm in diameter and 120 mm in height. On the other hand, specimens with 150 ± 1 mm in diameter, prepared in the laboratory, were compacted in a gyratory shear compacting press to the required density. The following subset of features was selected from the test results’ set for comparative purposes (Table 4):

The hypothesis stating that the specimens prepared in the laboratory differ substantially from specimens collected from the trial section was subjected to verification. The verification of the posed hypothesis was conducted with the assumed threshold error of 5%. The analysis featured the use of groups made of 4 ÷ 6 specimens. The verification of the presence of substantial difference was conducted using the ANOVA parametric test (F-test) and the Kruskal–Wallis non-parametric test (KW-H). The results of the comparative analysis are presented in Figure 5.

The ITSR results in Figure 5 demonstrate that there is no substantial difference between the average values in the “Type” category. Accordingly, the use of the mixed binder did not cause any substantial changes in the very important strength-related moisture resistance property, compared to the specimens prepared using the cement binder.

The results obtained from the KW-H test show no differences between UCS values, which might point to comparable axial compression strength in all specimens. An interesting observation was the high level of the ITS_DRY_ in the specimens prepared with the 5C mixed binder, similar to the results obtained for the set of specimens containing the cement binder. In relation to the IT-CY stiffness moduli, the results obtained were below 10,000 MPa, which was one of the assumptions of the project assuming a reduction in the recycled base stiffness. The UCS/ITS_DRY_ ratio in the specimens containing cement amounted to 3.3, whereas in the specimens collected from the trial section amounted to 2.2. Therefore, the trial section recycled mixture prepared from a mixed binder demonstrated lower stiffness but higher capacity to transfer tensile stress.

The comparative analysis was complemented by listing the results of feature designations for the specimens collected from the trial section and specimens prepared in the laboratory, containing the same 5C binder. The comparison included the mean value thanks to using the Student’s *t*-test. The results are tabulated in Table 5.

Based on the results presented in Table 5, V_m,_ and ITS_DRY_ were the only features that undoubtedly demonstrated any differences between the mean values of the specimens prepared in the laboratory and those collected from the pavement. The specimens collected from the trial section demonstrated V_m_ lower by 2.8% than specimens prepared in the laboratory. The same results were reported by researchers from the Gdansk University of Technology [42]. The ITS_DRY_ value of dry specimens collected from the trial section was 72 kPa higher than that of the specimens prepared in the laboratory. This fact was probably directly related to the presence of hydrated lime in the binder, which improved the cohesion of mastic [7]. This demonstrates that the CRM-FB_5C mixture prepared on the trial section was characterised by a more closed structure that the mixture prepared in the laboratory, which directly contributed to the improved ITS_DRY_ value. The value of V_m_ remains in the acceptable range of 8%÷15% in compliance with the requirements of the Polish Catalogue [43].

In terms of the UCS and ITSR parameters, no substantial differences between the mean values of the analysed specimen groups were recorded. Some specimens collected from the trial section achieved a higher UCS value than the specimens prepared in the laboratory, which will have a substantial positive effect on the recycled mixture’s strength. A similar lack of differences was recorded for stiffness IT-CY. Despite the lack of differences in the central measures, a greater range of results for specimens collected from the trial section was observed than in the case of specimens prepared in the laboratory. The possible reason for this could be the factor related to applying additional activities required for preparing specimens with adequate dimensions required for IT-CY testing with the use of a core. Figure 6 illustrates the relationships between the amount of individual components and the properties of the mixture containing the innovative binder.

The diagram in Figure 6 was prepared using the script in R-Studio. It follows from this diagram that the cement component most strongly correlates with stiffness at high temperatures, where the impact of bitumen is marginal. Its effect is weaker compared to stiffness at temperature below zero due to the high stiffness of the foamed bitumen at this temperature. Moreover, cement strongly affects the uniaxial compressive strength (UCS) and ITS_dry_. The Ca(OH)_2_ is most strongly correlated with air void content V_m_, and inversely correlated with the stiffness measured according to the IT-CY method. This is due to the fact that the presence of hydrated lime increases the compliance and susceptibility of the recycled mixture. The CBPD shows little correlation with water damage resistance (ITSR). Attention should be paid to the correlation trend. The interaction trends with opposite signs of each of the components suggests the presence of strong synergy between the binder components and selected properties of the recycled mixture.

### 3.2. Thermal Properties of Pavement Layers

The temperature distribution in the pavement cross-section was determined using the thermal coefficient λ. For this purpose, a temperature monitoring system was installed on the surface of the SMA-JENA layer (P1), at the interface between SMA-JENA and CRM-FB_5C (P2), and at the interface between the CRM-FB_5C and subgrade (P3) (Figure 4). The temperature distribution was recorded in the period between 17/04/2020 and 17/04/2021. Temperature data recorded at 240-min intervals were analysed. Temperature values at P1 and P3 were the boundary condition required for determining coefficient λ in differential Equation (1). The values of λ for the SMA-JENA and CRM-FB_5C layers were determined with back-calculations using the objective function. The convergence criterion was the minimum root mean square error (MSE) between the experimental results from P2 and the results obtained from the FEM numerical model. The MSE distribution obtained in the iterative process was described using a second-degree polynomial function. The sampling method was compliant with the rotatable response surface design [26]. The solution to the problem was to combine the coefficients λ for the CRM-FB and SMA-JENA layers in a manner that would enable the objective function to obtain the minimum mean square error. 

Another parameter related to heat transfer in the pavement cross-section was the linear thermal expansion coefficient α_T_. It is a quantity that determines the material’s ability to change its volume under the influence of temperature. The coefficient can be precisely determined under laboratory conditions or by using analytical formulae. The authors utilised the analytical method based on the Jones formula [44] and volumetric parameters of SMA-JENA and CRM-FB_5C mixtures. Parameters λ and α_T_ (Table 6) were used in further testing.

The fit error of λ at P2 was less than ε_SSE_ = 1.98 °C. Figure 7 shows a graphical representation of experimental and modelled temperature changes at P2 of the selected section with the CRM-FB_5C base course after data smoothing.

It must be noted that the best model fit to the experiment’s results was recorded between October 2020 and February 2021. This was probably due to the fact that the heat conductivity coefficient was determined as a constant value for the entire period. Nevertheless, it is known that its value must be determined for specific temperature and humidity ranges. The impact of a sudden temperature and humidity change can make the model’s forecast vary from the experimental values by a certain unknown error. Shewhart charts [46] appeared to be a useful tool. These charts are used to determine if a manufacturing process is in a state of control. The main assumption was based on the thesis that the MSE moving range should remain at a constant level. A substantial deviation from the mean value should point to the occurrence of a distorting factor. The results obtained by using the Shewhart charts are presented in Figure 8.

The temperature value recorded at the interface between the SMA-JENA and CRM-FB_5C layers was 11.5 °C on average (Figure 8b). Due to the high negative value of kurtosis, the most reliable mean value assessment was the median, which amounted to +9.2 °C and was slightly below the equivalent temperature in Poland, established to be +13 °C [43]. It is interesting that the most common temperature recorded at the layer interface was +0.1 °C. It follows from the above that the rheological properties of the recycled base course will depend greatly on the type of the mixed binder and foamed bitumen. Furthermore, it must be noted that the structure’s annual stiffness will probably be higher than the forecast stiffness, which assumes the temperature of +13 °C.

The distribution of the temperature difference between the numerically predicted and recorded values read with the P2 sensor shows that the highest discrepancies occurred in the period between April 2020 and September 2020 (Figure 8a). It was the period with the highest rainfall, when the biggest humidity difference between the top and bottom CRM-FB_5C layers was observed (above 9.4%). The average annual humidity difference was 8% (Figure 8c,d) and the coefficients λ for this value were most stable. The smallest temperature difference, 0.3 °C on average, between the predicted and measured data was recorded in the period from September to November (Figure 8a) when the overall humidity differences fluctuated within the confidence interval (5.8%; 9.4%). In this period, the highest humidity difference (>20%) occurred in the base course due to rain. On the other hand, it dropped to almost 0% in the period from January to March 2021. As demonstrated, base course humidity is not a constant factor and affects the value of the heat conductivity coefficient. Nevertheless, the observed discrepancy between the measured and predicted temperature distributions was minor and the satisfactory description of the temperature distribution across structural layers of the pavement allowed the correct implementation of the thermal expansion coefficient.

### 3.3. CRM-FB Base Layer Viscoelasticity

The range of controlled strains <50µε, as per PN-EN 12697-26 Annex D, warranted obtaining viscoelastic response of the material. The viscoelastic behaviour of the CRM-FB and SMA-JENA layers was described using a maximum of five systems of Maxwell model (GM) elements arranged in series. Parameters were identified by simultaneously minimising the difference between the experimental and modelled dynamic modulus and phase angle results. Table 7 summarises the results of the GM model parameter identification for the CRM-FB_5C, CRM-FB_cem base courses and SMA-JENA layer. 

The results presented in Table 7 demonstrate that the master curve model fit to the experimental data was performed with a low root mean square error of no more than 8%. Furthermore, the correlation coefficient was higher than 0.96. The highest dispersion of the results was recorded for the SMA-JENA layer on the basis of the root mean square error RMSE. The changes in the master curve simulated for the reference temperature +13 °C along with the fit assessment is presented in Figure 8.

It should be noted that the characteristics of each of the materials in the construction of the test section differ significantly from one another. The recycled base layer CRM-FB_cem exhibited the highest dynamic modulus and the lowest stress relaxation ability. There was, therefore, a high risk of fatigue-related or temperature-induced stresses. The CRM-FB_5C layer (5C binder consisting of CPBD, hydrated lime, and cement) was characterised by a much better ability to relax stresses, which, in the case of a recycled base course, is of crucial importance. The CRM-FB_5C curve (Figure 8a) shows that the layer achieved the lowest stiffness value. For comparison, the SMA-JENA mixture demonstrated a dynamic modulus in the range between those obtained by CRM-FB_cem and CRM-FB_5C, but had the highest stress relaxation rate. It is a desirable property for wearing and binder courses.

A much more complex viscoelastic profile of the SMA-JENA, CRM-FB_cem, and CRM-FB_5C recycled bases is shown by the black curve. A graphical representation of the black curve for the viscoelasticity of the specimens is given in Figure 9.

It can be seen in Figure 9 that the material in the CRM-FB_cem base course had the highest elasticity compared to other materials. The high dynamic modulus and the phase angle below 10° suggest its elastic–brittle nature. The use of the 5C binder reduced stiffness an increased viscous part of the dynamic modulus |E*|. Therefore, the asphalt mixture will be probably less sensitive to low temperature cracking as it has a higher elasticity reserve at low temperatures. At the same time, it meets one of the assumptions of the implemented composition optimisation process for the binder used (5C) [27]. Viscoelasticity of the SMA-JENA mixture depends on the binder cohesion. At low temperatures, SMA-JENA demonstrated the highest stiffness and the lowest phase angle, suggesting an elastic nature. On the other hand, at high temperatures, its viscous nature and the lowest dynamic modulus were dominant. Such diverse stiffness conditions of particular structure layers can be presented in a model form verified experimentally on the trial section.

### 3.4. Validation of Strains in the Trial Section 

The modelled and experimental strain results were compared for the horizontal strain E22 parallel to road axis and the horizontal displacement U3 cross-section. Before the basic numerical analysis was performed, the temperature distribution in the road cross-section was calculated in order to correctly estimate the change in material stiffness over time. The knowledge of the temperature was necessary for the additional displacement of the FEM mesh nodes due to the presence of additional strain or deformation dependent on the thermal expansion coefficient of the materials. The known pavement temperature and the form of the relaxation function (Table 5) allowed building a numerical model for the CRM-FB layer with an innovative binder, which was subjected to experimental validation. The adopted boundary conditions included the temperatures recorded at the trial section during the measurements on 4 September 2020 at 10:00. The set of temperature values read from the monitoring system and adopted based on the literature is presented in Table 8.

Figure 10 shows the calculated distribution of temperatures expressed in Celsius degrees recorded on 4 September 2020 on sample road measurement section made with the CRM-FB_5C recycled base course.

The shape function of the DC3D8 type (an 8-node linear heat transfer brick) was used to determine the temperature distribution. The obtained numerical set of results was implemented for further analysis of the deformation state as predefined temperature fields. The structure model was adopted as a 3D space with dimensions of 11.84 × 7.33 × 4.28 m (L–length × H–width × D–depth) and shown in Figure 11.

The original dimension of the FEM mesh element, 0.2 × 0.2 × 0.2 m, was reduced to 0.04 × 0.04 × 0.04 m for the interface. The H dimension of the JENA and CRM-FB_5C layers was split into four parts adding up to the thickness of each layer (volumetric elements). The C3D8R shape functions (8-node linear brick, reduced integration, hourglass control) were implemented as the shape function. The contact areas between vehicle wheels and the pavement were determined based on field measurements (recorded strain peaks) and commercial vehicle technical data. The test vehicle was loaded in a manner ensuring that the load generated on each axle corresponds to the data in Figure 11. Therefore, the rear wheel exerted the pressure of 850 kPa, whereas the front axle wheel exerted the pressure of 475 kPa. The adopted contact area was a circle 32 cm in diameter. This dimension was subjected to validation against the vertical strain at the SMA-JENA/CRM-FB_5C interface. The usefulness of the simulations was improved by including the interlayer slip effect resulting from manufacturing imperfections in the layer or induced by temperature effects. Ultimately, two solutions were adopted and the horizontal strain was read for both. The first solution assumed full friction connection denoted as E22_FEM full friction_. The second solution assumed the introduction of the friction coefficient φ = 0.8 between the SMA-JENA/CRM-FB_5C layers and the surface of the CRM-FB_5C/subgrade. This solution was denoted as E22_FEM penalty (f.c. = 0.8)_ to represent the strain range in which the pavement experienced a substantial loss of friction connection [47]. The experimental horizontal strain E22_exp was read using the data from the sensors placed parallel to the road axis at the vehicle load point with the stress of q1 = 850 kPa. Additional measurements with the FWD deflectometer allowed determining the equivalent minimum elasticity modulus of the subgrade, which was 156 MPa (5% probability distribution quartile). The simulation results are presented in Figure 12.

It is necessary to remember that the changes in the simulated strain over time were determined with the use of the stiffness modulus results during the recycled base’s relaxation phenomenon tests conducted at the laboratory. The results presented in Figure 12 demonstrated a certain discrepancy in the first 200 s. This was due to the fact that it was not possible to conduct the strain tests immediately when the vehicle’s wheels approached the monitoring area or immediately take readings from all optic-fibre sensors. This strain range can be presented theoretically as an extrapolation based on the results from subsequent reading times when the vehicle’s position was stable.

When comparing the results of the two numerical simulation cases with the experimental results for the CRM-FB mixture (Figure 12), it must be stated that the structure’s response pointed to the presence of a probable interlayer slip. The observed circumstances suggest a certain loss in connection when compared to the solution with a full friction connection, however its value remained acceptable [47]. The experimental values of the horizontal strain E22 were more similar to the results of the simulation, which assumed a slight interlayer slip. The interlayer slip is a common random issue in road construction, affecting the pavement fatigue life. According to Prof. Jaskuła [42], a minor cohesion-related slip is always present. For this reason, in numerical analyses a slight drop in friction between the layers (φ = 0.8) was taken into account. The slip was probably caused by different values of heat conductivity coefficient and the related thermal expansion coefficients. Different viscoelastic characteristics also cause a certain slight slip between the pavement layers vehicle load. 

Figure 12 also includes the result for the CRM-FB recycled base used with the adopted stiffness modulus E = 1500 MPa and for the SMA-JENA mixture with the stiffness of E = 7300 MPa. This was a predefined value specified in the Polish catalogue for the load time of 0.02 s and the reference temperature of 13 °C [43]. This variant, denoted in Figure 12 as E22_CAT full friction_, assumed a full friction connection. The temperature on the road slightly exceeded the reference temperature. The simulated strain solution is presented as a constant strain value because the elastic model applies in the catalogue. A large discrepancy must be noted with respect to the actual strain when comparing the experimental results with the model result (E = 1500 MPa). The strain values (E22_CAT full friction_ = 143 × 10^−6^ m/m) coincide with the experiment only after approximately 500 s and only with the assumption that stiffness as per the catalogue does not change. Therefore, it must be emphasised that the actual potential of the recycled mixture with a three-component binder was substantially greater than suggested in the Polish catalogue. During a 1 s load, the strain under the vehicle’s rear axle wheel amounted to E22 = 92.1 × 10^−6^ m/m for CRM-FB_5C, provided that the variant assumed was the one in which the interlayer slip occurred. The value was 36% lower when compared to the catalogue assumptions (for newly-designed mixtures). The catalogue does not assume the slip and the recycled layer thickness is E = 1500 MPa, whereas for SMA-JENA, E = 7300 MPa at a temperature of 13 °C and loading time of 0.02 s. The layer modules obtained (Table 7) were markedly higher for the same loading time, which resulted in lower strain values. 

Another quantity subjected to measurement validation was the vertical displacement U3 recorded under the vehicle rear wheel. The experimental results were compared to two numerical variants taking into consideration the friction connection as in the case of the horizontal strain analysis. The vertical displacement and horizontal strain results are presented in Figure 13.

Analysis of Figure 13a shows that, unlike in the horizontal strain analysis, the results of the maximum displacement under the pavement with the CRM-FB_5C mixture are more convergent with the numerical solution in which no interlayer slip occurs. The recorded discrepancies could result from placing the deflection sensor under the recycled base layer instead of embedding it in the layer. Phenomena that occur in the subgrade could cause certain changes in the displacement path. Nevertheless, the experimental measurement values were comparable with the simulated results. 

The values of strain E22 (Figure 13b) recorded by the longitudinal optic-fibre sensors are closer to the result with the assumption of certain interlayer slip (μ = 0.8). The strain results and strain changes along the sensor measurement length were consistent. It can thus be stated that the results of mechanical properties entered into the numerical program in the form of physical models represent the pavement behaviour very high accuracy. The conclusions drawn about the quality of the CRM-FB_5C recycled base are confirmed by field data.

## 4. Conclusions

The paper focuses on the assessment of the viscoelastic properties of recycled mixtures placed in a trial section and validation of the assumed material constants against the field data (trial section responses). Heat conductivity coefficient was used to determine the correct temperature distribution in the pavement structure cross-section. The following conclusions were formulated based on the tests and analyses:
The strain and displacement data from the monitoring system confirmed the high effectiveness of the applied model based on the generalised Maxwell model used for describing the relaxation of the recycled mixture specimens. The similarity between the field data (trial section) and numerical simulations indicates that these are no signs of damage in the recycled base;Monitoring temperature and humidity allowed determining the heat conductivity coefficient, which, in turn, helped to estimate the temperature distribution, hence, strain in the pavement structure (trial section);Heat conductivity coefficient variation could be the cause of the slight interlayer slip in the early period of pavement service;Humidity levels in the recycled base course contributed to the heat conductivity coefficient fluctuations. Nevertheless, the fit error between the temperature measured and calculated during 12 months did not exceed 1.98 °C on average;A substantial difference in the ITS_DRY_ and V_m_ results was observed between the field and laboratory specimens. The specimens collected from the pavement had a more closed structure and higher ITS_DRY_ values in comparison to those prepared in the laboratory;The UCS/ITS_DRY_ ratio for the CRM-FB_cem specimens amounted to 3.3 compared to 2.2 for the CRM-FB_5C specimens collected from the trial section. Therefore, the trial section recycled mixture prepared from the mixed binder had lower stiffness but higher indirect tensile strength ITS_DRY_;No substantial difference in the ITSR characteristic was observed between the CRM-FB_cem and CRM-FB_5C specimens or between the specimens collected from the trial section and those prepared in the laboratory;The CRM-FB_5C mixture was more viscous and had a higher relaxation capacity than the CRM-FB_cem mixture. Accordingly, the probability of thermally-induced crack formation is lower in the CRM-FB_5C specimens;The pavement response comparison based on numerical and simulated data revealed a 36% lower strain than that simulated based on the reference material data provided in the Polish catalogue. This conclusion is especially important in terms of the road structure increased fatigue life;There is a strong correlation between cement and stiffness, strength, and indirect tensile strength. The presence of hydrated lime was strongly associated with V_m_ and the CBPD with the ITSR. In addition, strong interaction relationships were observed between the components of the hydraulic binder and the obtained material quality and selected physical and mechanical properties.

## Figures and Tables

**Figure 1 materials-14-05925-f001:**
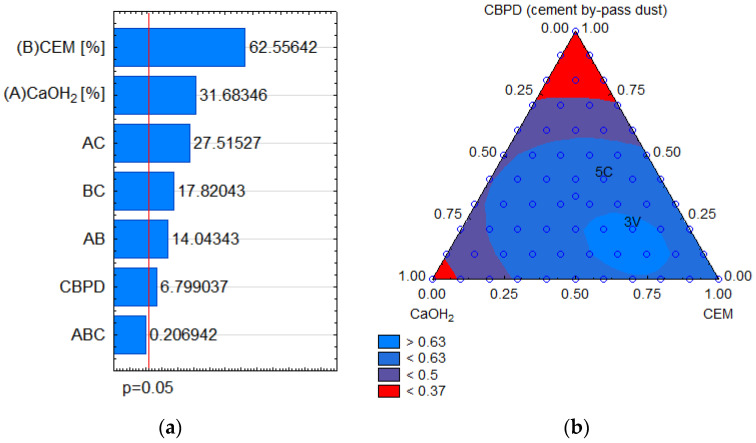
Three-component road binder optimisation result: (**a**) Pareto chart; (**b**) desirability function result of binder quality.

**Figure 2 materials-14-05925-f002:**
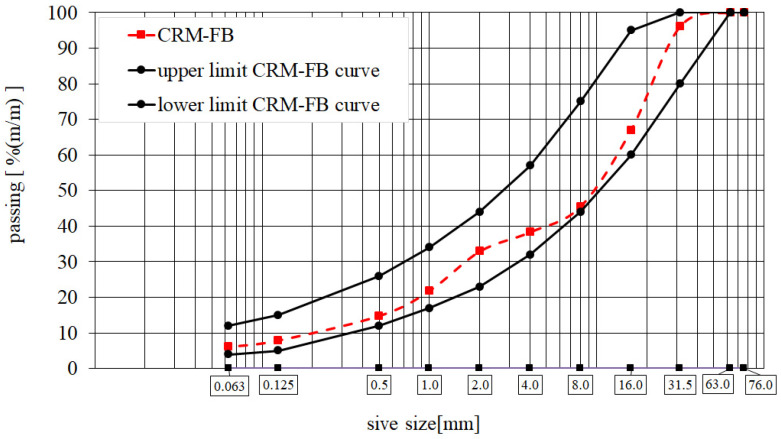
CRM-FB gradation curve.

**Figure 3 materials-14-05925-f003:**
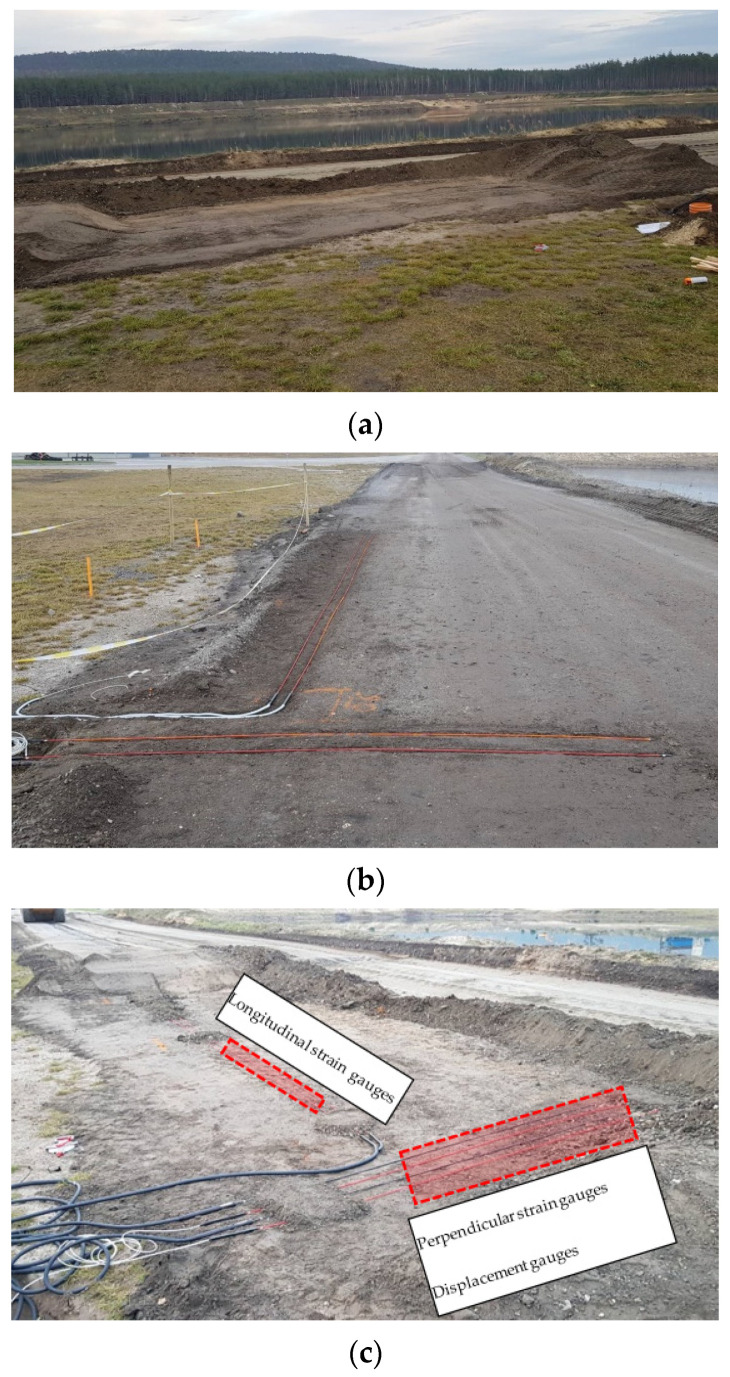
Trial section construction process: (**a**) subgrade preparation for the installation of sensors in the bottom part of the CRM-FB_5C base course; (**b**) sensor installation under the CRM-FB_5C base layer; (**c**) sensor installation under the wearing/binder layer of the JENA type SMA16 mixture.

**Figure 4 materials-14-05925-f004:**
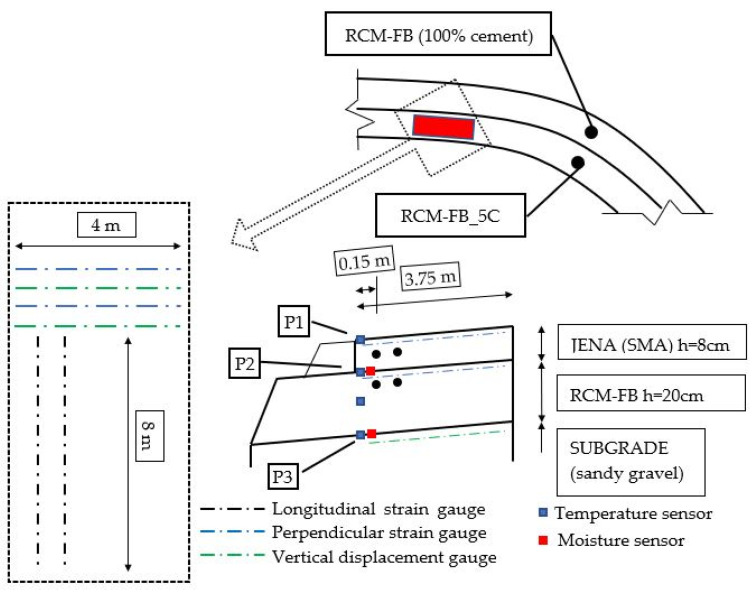
Schematic diagram of the monitoring sensor arrangement.

**Figure 5 materials-14-05925-f005:**
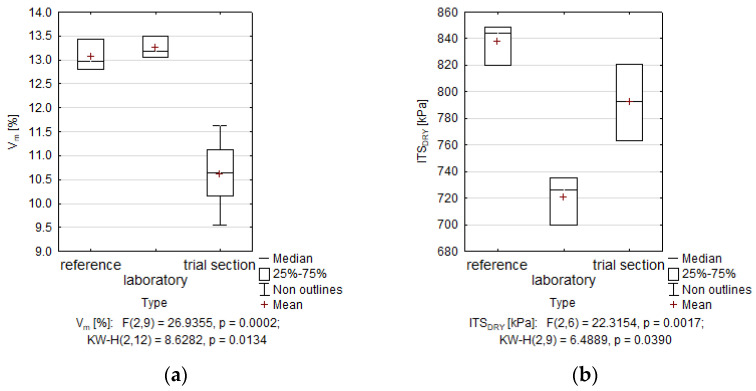
Comparison of the properties results for specimens prepared in the laboratory and collected from the trial section with the results for specimens prepared using solely with the reference cement binder: (**a**) Void content V_m_; (**b**) Indirect tensile strength ITSdry; (**c**) uniaxial compression strength UCS; (**d**) indirect tensile strength ratio ITSR; (**e**) indirect strength stiffness modulus at 5 °C deg. IT-CY, (**f**) indirect strength stiffness modulus at +13 °C deg. IT-CY.

**Figure 6 materials-14-05925-f006:**
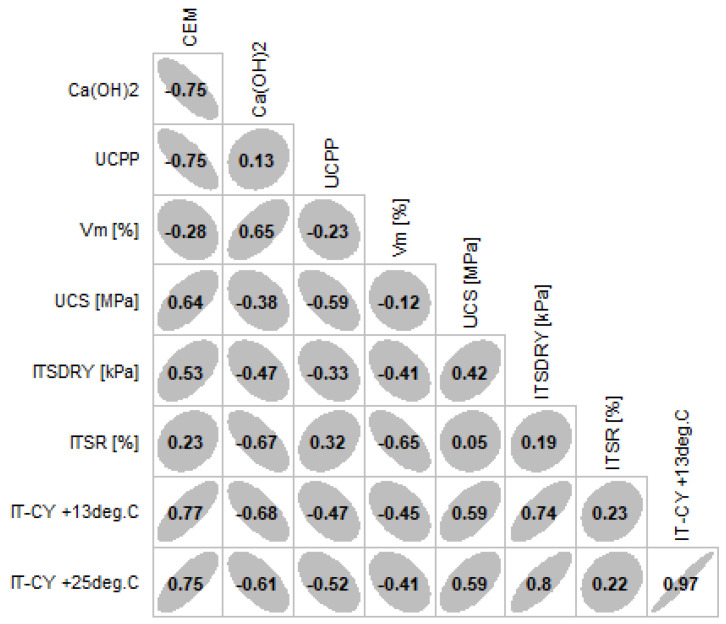
Correlation between selected CRM-FB layer and amount of innovative binder constituents.

**Figure 7 materials-14-05925-f007:**
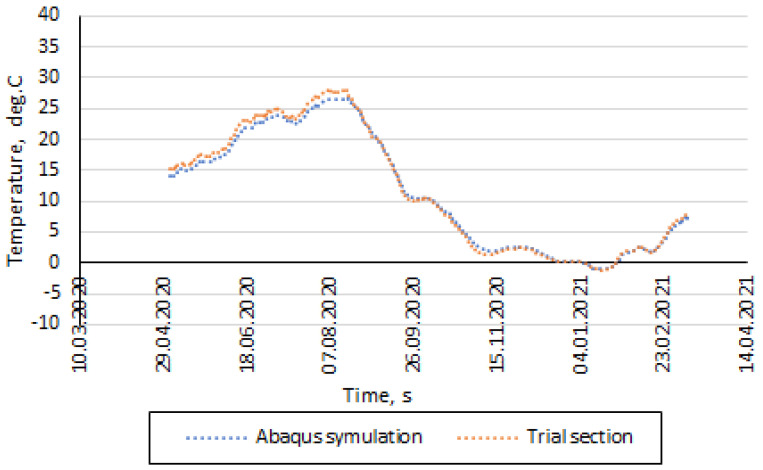
Temperature distribution at the interface between the SMA-JENA and CRM-FB_5C layers, recorded by the P2 sensor, after data smoothing.

**Figure 8 materials-14-05925-f008:**
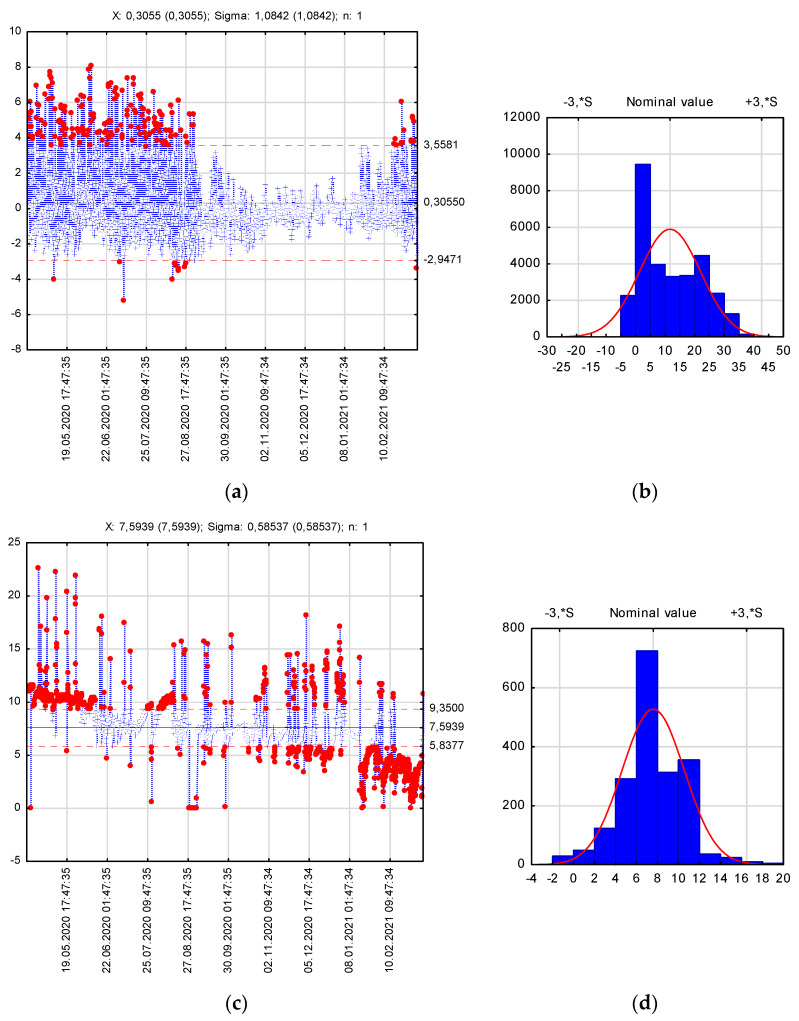
Shewhart charts for monitoring data forecast: (**a**) temperature MSE moving range; (**b**) temperature probability range; (**c**) humidity difference between the top and bottom surface of the CRM-FB_5C layer; (**d**) probability range for humidity difference between the top and bottom surface of the CRM-FB_5C layer.

**Figure 9 materials-14-05925-f009:**
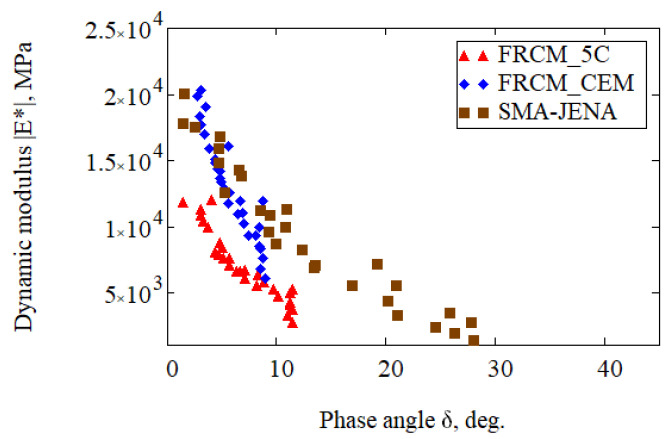
Black curve for the trial section materials.

**Figure 10 materials-14-05925-f010:**
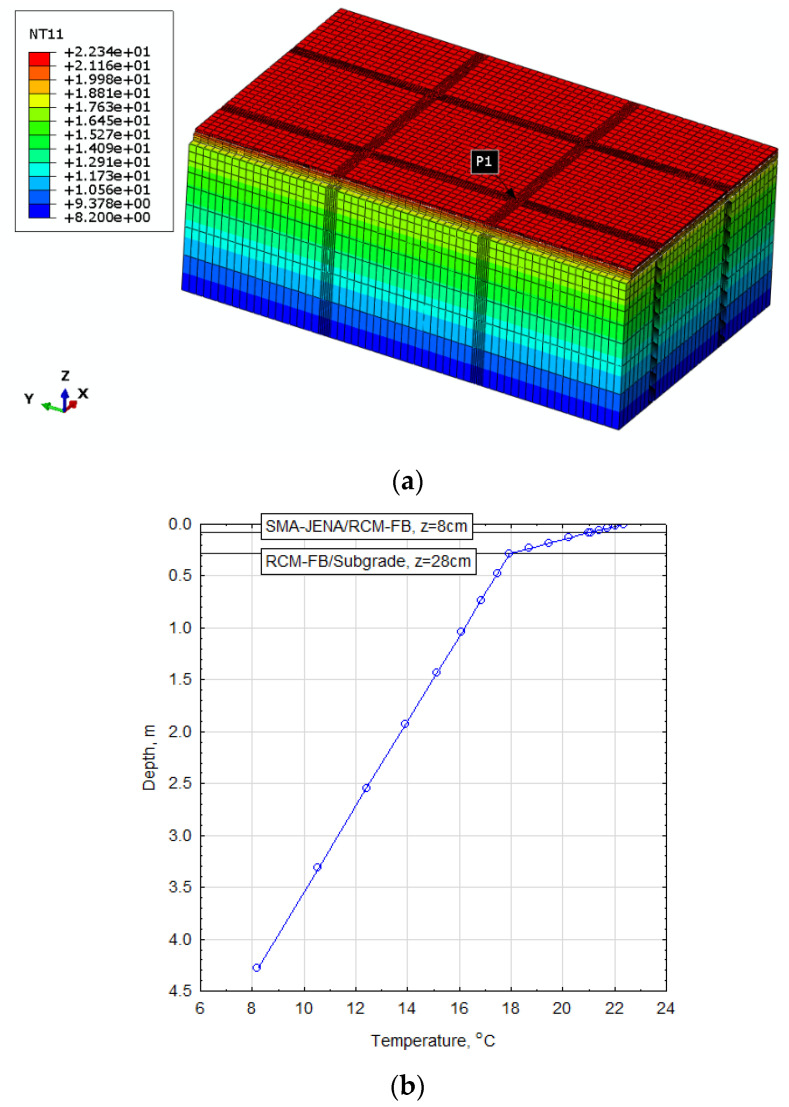
Temperature distribution in the cross-section of the road constructed with the CRM-FB_5C base: (**a**) 3D view of temperature distribution; (**b**) cross-section at point P1 (z-coordinate).

**Figure 11 materials-14-05925-f011:**
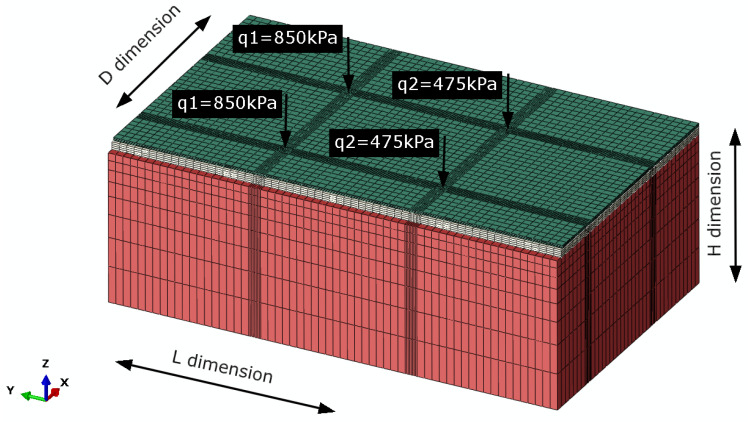
Loading scheme for the structural layers of the trial section; with the assigned FEM mesh.

**Figure 12 materials-14-05925-f012:**
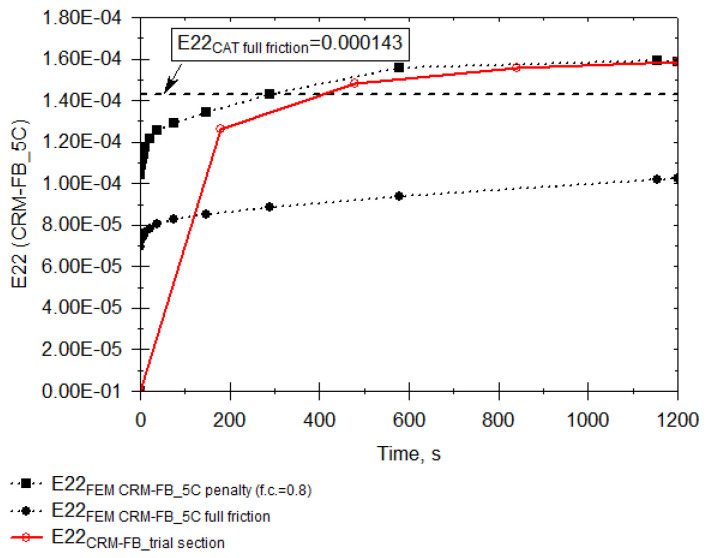
Distribution of the horizontal strain E22 in a time function under the vehicle’s wheel, underneath the CRM-FB_5C layer.

**Figure 13 materials-14-05925-f013:**
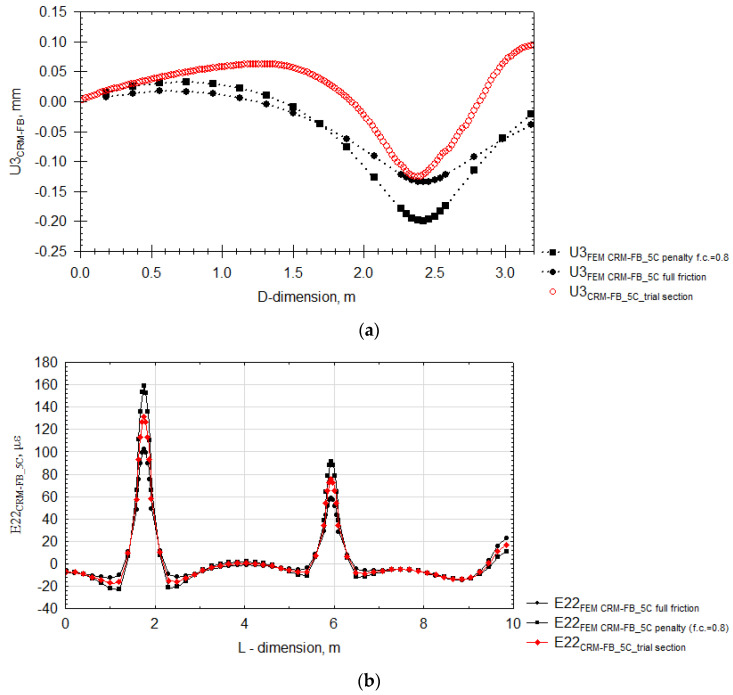
Changes in strain and displacement under the CRM-FB_5C recycled base after 1200 s: (**a**) vertical displacement U3 versus D dimension; (**b**) strain E22 versus L-dimension.

**Table 1 materials-14-05925-t001:** Quantity of components of a universal binder based on the mixture design with constraints.

Component Combinations	Components		
	Hydrated Lime (HL)	Cement (CEM)	Cement By-Pass Dust (CBPD)
7 C(2)	1/3	1/3	1/3
6 C(1)	2/5	2/5	1/5
5 C(1)	2/5	1/5	2/5
1 V	1/5	1/5	3/5
4 C(1)	1/5	2/5	2/5
3 V	1/5	3/5	1/5
2 V	3/5	1/5	1/5

**Table 2 materials-14-05925-t002:** Phase composition of raw materials (%).

CEM I 32.5R		CBPD		Ca(OH)_2_	
C3S (alite)	65.3	CaO	43.6	Portlandite	97.4
C2S (belite)	10.0	Sylvine	16.7	Calcite	2.6
C4AF	4.4	C2S (belite)	34.5		
C3A	9.3	Calcite	5.1		
Arcanite	1.3				
Gypsum	0.9				
Calcite	7.7				
Quartz	1.0				

**Table 3 materials-14-05925-t003:** Test results of the bitumen.

Properties		Road Bitumen 70/100	Standard
Penetration at a temperature of 25 °C, 0.1 mm		84 ± 3.9	EN 1426
Softening point R&B, °C		47 ± 1.1	EN 1427
Breaking point according to Fraass, °C		−18 ± 0.5	EN 12593
Bending Beam Rheometer (BBR)	S(at 60 °C) = 300 MPa	−19.2 ± 0.5	EN 14771
	m(at 60 °C) = 0.3	−17.9 ± 0.6	
Plasticity range (T_R&B_-T_Fraass_)		65 ± 1.2	EN 14023
Viscosity at 60 °C, Pas		151 ± 5.4	ASTM D 4402
Viscosity at 90 °C, Pas		8.3 ± 0.3	
Viscosity at 135 °C, Pas		0.38 ± 0.01	
Viscosity at 150 °C, Pas		0.14 ± 0.007	

**Table 4 materials-14-05925-t004:** Selected physical and mechanical features of the recycled base.

Feature	Unit	Standard
Void content, V_m_	%	EN 12697-30
Indirect tensile strength, ITS	kPa	EN 12697-23
Uniaxial compressive strength, UCS	MPa	EN 13286-41
Water and frost resistance, ITSR	%	EN 12697-12
Stiffness at +5 °C, IT-CY +5 °C	MPa	EN 12697-26, Annex C
Stiffness at +13 °C, IT-CY +13 °C	MPa	EN 12697-26, Annex C

**Table 5 materials-14-05925-t005:** Student *t*-test for comparisons of the mean values for specimens prepared in the laboratory and specimens collected from the trial section for the CRM-FB base with the 5C binder.

Feature	Unit	Group 1: Laboratory Group 2: Trial Section			
		Mean Laboratory	Mean Trial Section	*p*-Value (Equal Variance)	*p*-Value (Unequal Variance)
V_m_	[%]	13.25	10.62	0.000772	0.000174
UCS	[MPa]	2.07	2.23	0.382319	0.393148
ITS_DRY_	[MPa]	720.33	792.33	0.021901	0.029490
ITSR	[%]	67.37	63.40	0.152091	0.166845
IT-CY + 5 °C	[MPa]	7512.25	8985.30	0.136241	0.126831
IT-CY + 13 °C	[MPa]	7045.50	8011.80	0.176142	0.157460

**Table 6 materials-14-05925-t006:** Trial section thermal parameters.

Layer	Heat Conductivity Coefficient λ, W/(m·°C)	Linear Thermal Expansion Coefficient α_T,_ 1/°C
SMA-JENA	0.1	2.7 × 10^−^^7^
CRM-FB_5C	2.4	1.8 × 10^−^^5^
Subgrade	0.58 [45]	5 × 10^−^^6^

**Table 7 materials-14-05925-t007:** Master curve parameters of SMA-JENA and CRM-FB on the basis of generalised Maxwell model (GM) at a reference temperature of 13 °C.

Technology Type	Simple Maxwell Model Parameters		Factor α_T_ (WLF Formula)	
	G_i_ [-]	τ_i_ [s]	C_1_	C_2_
**CRM-FB recycled base (100% cement)**	G_1_ = 0.2 G_2_ = 0.2 G_3_ = 0.2 G_4_ = 0.2 G_5_ = 0.2	τ_1_ = 0.00177 τ_2_ = 0.14122 τ_3_ = 12.7251 τ_4_ = 2035.12 τ_5_ = 52162.1	−17.0	163.2
	G_o_ = 7735 MPa			
	R^2^ = 0.96; RMSE = 4.8%			
**CRM-FB_5C recycled base (5C binder)**	G_1_ = 0.229 G_2_ = 0.193 G_3_ = 0.193 G_4_ = 0.193 G_5_ = 0.193	τ_1_ = 0.00097 τ_2_ = 0.08296 τ_3_ = 5.06899 τ_4_ = 213.756 τ_5_ = 12,007.6	−7.1	67.1
	G_o_ = 4393 MPa			
	R^2^ = 0.97; RMSE = 5.3%			
**SMA-JENA wear/binding layer**	G_1_ = 0.252 G_2_ = 0.252 G_3_ = 0.222 G_4_ = 0.177 G_5_ = 0.096	τ_1_ = 0.00085 τ_2_ = 0.06414 τ_3_ = 2.79957 τ_4_ = 110.362 τ_5_ = 6066.03	−32.4	229.8
	G_o_ = 6946 MPa			
	R^2^ = 0.99; RMSE = 7.8%			

**Table 8 materials-14-05925-t008:** Temperatures read from the monitoring system (boundary conditions).

Location (Depth, cm)	Temperature °C
SMA-JENA surface (0 cm)	22.34
SMA-JENA/CRM-FB_5C interface (8 cm)	21.10
CRM-FB_5C/subgrade interface (28 cm)	17.95
Subgrade (500 cm) [21]	8.2

## Data Availability

Data available on request from the corresponding author.

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
