# Peer review of "Thermal Analysis-Based Field Validation of the Deformation of a Recycled Base Course Made with Innovative Road Binder"

_materials, 2021, doi:10.3390/ma14205925_

Round 1
Reviewer 1 Report
Line article number (LAN) 16-18 ....The paper presents an analysis of the results of strain in the FRCM recycled base with an innovative three-component road binder and foamed bitumen (recycled cold mixture with foamed bitumen)…I recommend considering an explanation of the FRCM (recycled cold mixture with foamed bitumen) designation to exactly match the abbreviation used.
LAN 56-58…It is necessary to remember that the presence of each of the components in the binder plays the role of the foamed bitumen’s dispersion catalyst or bitumen emulsion degradation, resulting in substantially improved mechanical parameters of recycled mixtures…It would be appropriate to specifically mention which mechanical properties are occurring.
LAN 109… The sum of the mixture’s components could not be greater than 100 %...Of course, we can only agree with this statement, but it must also be true that the amount must not be less than 100%...For this reason, I recommend using a fractional expression of 1/3 ratios in Table 1.
LAN 120… Figure 1. Hydraulic binder optimisation result…I recommend adding an explanation of the abbreviation CPBG to the text or the legend of the figure and aligning the designation of cement (C or CEM) with Table 1.
LAN 149… Figure 2. FRCM granulation curve. It is necessary to correct the description of the x-axis on the sieve size.
LAN 173 and 183…Figure 3 and 4. Typos need to be corrected: Perpendiducinal gauges, Perpendidunical strain gauge.
LAN 206… λ – heat conductivity coefficient, W/(m·ºC)…It is common to state the physical unit λ in the form (W.m-1.K-1).
LAN 396-397… The value was slightly below the equivalent temperature in Poland, determined to be equal to +13ºC [45]. The air temperature is significantly dependent on the altitude of the assessed locality… Within Slovakia, the average air temperature determined for the period 1970 to 2020 in the inhabited areas (altitude 100 to 1000 m) ranges from 5 to 10.5 ºC. During this period, an increase in average annual temperatures of 2 ºC was found. Does the above temperature “equivalent temperature in Poland, determined to be equal to +13ºC” take into account the significant increase in temperatures in Central Europe over the last 50 years?
Figure 10. Temperature distribution in the cross-section of a road with the FRCM_5C base. From the point of view of increasing the informative value of Figure 10, it would be appropriate to add numerical advisors of the monitored depth of the road structure.
LAN 478-482…The primary numerical analysis was preceded by the numerical calculation of the temperature distribution in the road’s cross-section to correctly estimate the stiffness of the materials used in the road’s cross-section and to additionally shift the FEM mesh nodes due to the occurrence of an additional strain which depended on the materials’ thermal expansion coefficient... In what way, for example numerically, empirically, graphically, was the correct estimate of the stiffness of the materials used in the road’s cross-section?
LAN 534-537…When comparing the results of the two numerical simulation cases with the experimental results for the FRCM mixture (Figure 12), it must be stated that the structure’s response pointed to the presence of a probable interlayer slip…I recommend briefly adding, by which you explain “the presence of a probable interlayer slip” and whether at this stage of the solution it is possible to outline its scope.
LAN 599-602…The temperature and humidity monitoring allowed for identifying the heat conductivity coefficient, which in turn allowed for estimating the temperature distribution in the pavement structure’s trial section and, in consequence, its strain condition… The value given ”identifying the heat conductivity coefficient” is convergent with the value in Polish Catalogue.
LAN 611-613… The samples collected from the pavement had a more closed structure and higher ITSDRY values in comparison to the samples prepared in the laboratory… How does it explain the differences in the “closedness” of the structure in situ and laboratory samples? The differences in question could be due to the compaction of the samples during construction or operation of the pavement of interest?
LAN 625-630… The comparison of the structure’s responses using the data derived from the monitoring system and the numerical modelling process indicated that the strain is 36 % smaller than the result obtained based on the strain condition simulation using the reference material data provided in the Polish Catalogue. This conclusion is especially important in terms of the road structure’s increased fatigue life…It is necessary to add on the basis of which compared the mechanical quantity the mentioned difference of 36% was identified. It is also necessary to refer to the Polish Catalogue in this section.
Overall, I rate the article as excellent, I would like to highlight the amount of research work on isomorphic and also homomorphic models of semi-rigid pavement construction with innovative materials. Assessed scientific contribution, after removing minor deficiencies, I recommend for publication.
Author Response
We have made efforts to improve the style of the English language by sending the manuscript for professional proofreading. The comments submitted by the reviewer are marked in green. However, additional information that requires additional correction were yellow and gray colored.
Line article number (LAN) 16-18 ....The paper presents an analysis of the results of strain in the FRCM recycled base with an innovative three-component road binder and foamed bitumen (recycled cold mixture with foamed bitumen)…I recommend considering an explanation of the FRCM (recycled cold mixture with foamed bitumen) designation to exactly match the abbreviation used.
Answer: The recycled mixture name has been changed to RCM-FB in the manuscript
LAN 56-58…It is necessary to remember that the presence of each of the components in the binder plays the role of the foamed bitumen’s dispersion catalyst or bitumen emulsion degradation, resulting in substantially improved mechanical parameters of recycled mixtures…It would be appropriate to specifically mention which mechanical properties are occurring.
Answer: Appropriate information was added to manuscript
LAN 109… The sum of the mixture’s components could not be greater than 100 %...Of course, we can only agree with this statement, but it must also be true that the amount must not be less than 100%...For this reason, I recommend using a fractional expression of 1/3 ratios in Table 1.
Answer: The comment was included in the manuscript.
LAN 120… Figure 1. Hydraulic binder optimisation result…I recommend adding an explanation of the abbreviation CPBG to the text or the legend of the figure and aligning the designation of cement (C or CEM) with Table 1.
Answer: The comment was included in the manuscript.
LAN 149… Figure 2. FRCM granulation curve. It is necessary to correct the description of the x-axis on the sieve size.
Answer: The comment was included in the manuscript.
LAN 173 and 183…Figure 3 and 4. Typos need to be corrected: Perpendiducinal gauges, Perpendidunical strain gauge.
Answer: The comment was included in the manuscript.
LAN 206… λ – heat conductivity coefficient, W/(m·ºC)…It is common to state the physical unit λ in the form (W.m-1.K-1).
Answer: The comment was included in the manuscript.
LAN 396-397… The value was slightly below the equivalent temperature in Poland, determined to be equal to +13ºC [45]. The air temperature is significantly dependent on the altitude of the assessed locality… Within Slovakia, the average air temperature determined for the period 1970 to 2020 in the inhabited areas (altitude 100 to 1000 m) ranges from 5 to 10.5 ºC. During this period, an increase in average annual temperatures of 2 ºC was found. Does the above temperature “equivalent temperature in Poland, determined to be equal to +13ºC” take into account the significant increase in temperatures in Central Europe over the last 50 years?
Answer: Thank you for a valuable comments.The equivalent temperature was determined by the Gdańsk University of Technology at the request of the General Directorate of Roads and Motorways in 2010-2014. On the basis of this reports, the authors mainly used the meteorological data. In our case, we had sensors at our disposal and we did direct measurement. Unfortunately, we do not know the effects of temperature fluctuations, so the average temperature may be slightly higher. Nevertheless, the monitoring will be carried out in the next 10 years, which will certainly have an impact on the verification of the current state of affairs in Poland. I would like to add that the latest research carried out under the RID program (Prof. Marek Pszczoła from the Gdańsk University of Technology) made forecasts (due to climate warming) stating that the equivalent would be increased to +15deg.C.
Figure 10. Temperature distribution in the cross-section of a road with the FRCM_5C base. From the point of view of increasing the informative value of Figure 10, it would be appropriate to add numerical advisors of the monitored depth of the road structure.
Answer: The comment was included in the manuscript as a result of Figure 10b addition.
LAN 478-482…The primary numerical analysis was preceded by the numerical calculation of the temperature distribution in the road’s cross-section to correctly estimate the stiffness of the materials used in the road’s cross-section and to additionally shift the FEM mesh nodes due to the occurrence of an additional strain which depended on the materials’ thermal expansion coefficient... In what way, for example numerically, empirically, graphically, was the correct estimate of the stiffness of the materials used in the road’s cross-section?
Answer: An appropriate supplementation was added to the manuscript. Nevertheless, it was essential to know the relaxation function. Due to the fact that the deformation of the viscoelastic materials depends on the temperature and the loading time, it was also required to have an exact temperature distribution in the cross-section. The specified relaxation function (physical material model) was used to build the FEM model and its correctness was confirmed by the deformation results from the monitoring. Moreover, the stiffness values of the samples taken from the pavement did not differ significantly from the results of the samples prepared in the laboratory. Thus, it could be concluded that the model has corresponded to reality.
LAN 534-537…When comparing the results of the two numerical simulation cases with the experimental results for the FRCM mixture (Figure 12), it must be stated that the structure’s response pointed to the presence of a probable interlayer slip…I recommend briefly adding, by which you explain “the presence of a probable interlayer slip” and whether at this stage of the solution it is possible to outline its scope.
Answer: During designing a new pavement structure in Poland, a full interlayer adhesion is assumed. Nevertheless, imperfections in time of layer placing always occur. They affect the interlayer connection. The quality of the connection is assessed in the Leutner apparatus. The tests, which were carried out in the laboratory, have shown that the shear stress between layers was greater than 1.0 MPa. Therefore it should be assumed that there is an interlayer bond of a cohesive nature, i.e. in the range of 80-100%. In order to capture the range of potential errors, a decrease in the connection was assumed and expressed as a friction coefficient of 0.8 (we did of such assumption on the basis of the experience of Prof. Piotr Jaskuła from the Gdańsk University of Technology). A relevant comment was added to the text of the manuscript.
LAN 599-602…The temperature and humidity monitoring allowed for identifying the heat conductivity coefficient, which in turn allowed for estimating the temperature distribution in the pavement structure’s trial section and, in consequence, its strain condition… The value given ”identifying the heat conductivity coefficient” is convergent with the value in Polish Catalogue.
Answer: This value of the temperature conductivity coefficient was experimentally estimated using the backward method. Unfortunately, there is no data in the Polish catalog that would approximate this value. The adopted value of the thermal conductivity coefficient was estimated for a given case of the material. Nevertheless, it falls between the range of variability of similar materials of the pavement layers found in literature.
LAN 611-613… The samples collected from the pavement had a more closed structure and higher ITSDRY values in comparison to the samples prepared in the laboratory… How does it explain the differences in the “closedness” of the structure in situ and laboratory samples? The differences in question could be due to the compaction of the samples during construction or operation of the pavement of interest?
Answer: The sample compaction process in the laboratory was carried out using a gyratory press. The recycled layer was made in field conditions, where the pressure of the roller and the resistance of the subgrade were different than in the laboratory. Furthermore, mixing was taking place in situ so the void content locally could have been different. Nevertheless, its average value of 10.5% was within the recommended range (from 8% to 15%). The other researchers reported a similar situation in the article(Jaskula, P.; Ejsmont, J.; Stienss, M.; Ronowski, G.; Szydlowski, C.; Swieczko-Zurek, B.; Rys, D. Initial Field Validation of Poroelastic Pavement Made with Crumb Rubber, Mineral Aggregate and Highly Polymer-Modified Bitumen. Materials 2020, 13, 1339, doi:10.3390/ma13061339.)
LAN 625-630… The comparison of the structure’s responses using the data derived from the monitoring system and the numerical modelling process indicated that the strain is 36 % smaller than the result obtained based on the strain condition simulation using the reference material data provided in the Polish Catalogue. This conclusion is especially important in terms of the road structure’s increased fatigue life…It is necessary to add on the basis of which compared the mechanical quantity the mentioned difference of 36% was identified. It is also necessary to refer to the Polish Catalogue in this section.
Answer: The reported difference in deformations resulted from the fact that the stiffness modulus values included in Polish Catalog turned out to be significantly lower than in the designed material and presumably didn’t depend on time. Even with the assumption of imperfection related to inter-layer slip, the response of the structure manifested itself with smaller deformations than taking into account the catalog data. With a loading time of 1s (vehicle moving approx. 1km / h), the stiffness of the structure using the innovative material still had a higher value than the level established in the Polish Catalog.
Reviewer 2 Report
Dear authors.
During my review of the paper titled "Experimental verification of strain in a recycled base containing an innovative road binder taking into account the layers’ thermal analyses" I realised that the article is not really about materials. It is mainly about road structure in which one layer is made of recycled cold mixture with foamed bitumen.
Thus I highly recommend to the authors to extend the Materials and Methods part. They should give all necessary data about used basic materials (constituents), they should describe used test methods at least briefly, although the standard test methods were used. I am missing clear presentation of the test results and discussion about obtained test results that would address the relation between measured properties and material/layer composition.
You can find additional comments in the attached document.

Author Response
Thank you for a valuable comments. That is true we paid a great attention on validation of innovative binder influence in recycled base. It was a final stage of realized scientific project. We made a lot of analyses on relation between components and recycled layer properties. For this a computer program has been elaborated. Nevertheless our intention was to show how this material behave in real conditions because laboratory test has not allowed as acquire a knowledge how potential placing process imperfection affected layer behavior. Generally we have based on laboratory data giving mechanical description to materials. So generally we tried to show hos this innovative materials behave in real road structure keeping in mind that article concerned a new material. Results exceeded our expectations.
We accord with reviewer that basic information on material/binder behavior should be added. So additional figure 1a and figure 6 were incorporated into text of manuscript. The both dealt with relation between binder constituents and utility function as well as recycled mixture features. We also added a short description of properties used in analyses. Moreover we sent this manuscript for professional correction to eliminate mistakes as much as possible. All correction did in compliance to reviewer comments were yellow colored. Additional information that requires additional correction were green and gray colored.